# *lmo4a* Contributes to Zebrafish Inner Ear and Vestibular Development via Regulation of the Bmp Pathway

**DOI:** 10.3390/genes14071371

**Published:** 2023-06-28

**Authors:** Le Sun, Lu Ping, Ruzhen Gao, Bo Zhang, Xiaowei Chen

**Affiliations:** 1Department of Otolaryngology, Peking Union Medical College Hospital, Chinese Academy of Medical Sciences and Peking Union Medical College, #1 Shuaifuyuan, Dongcheng District, Beijing 100730, China; pumc_sunle@student.pumc.edu.cn; 2Chinese Academy of Medical Sciences and Peking Union Medical College, #9 Dongdan Santiao, Dongcheng District, Beijing 100050, China; lu_ping@student.pumc.edu.cn; 3Department of Clinical Laboratory, Peking Union Medical College Hospital, Chinese Academy of Medical Sciences and Peking Union Medical College, Beijing 100730, China; gao_rz@pumch.com; 4Key Laboratory of Cell Proliferation and Differentiation of the Ministry of Education, College of Life Sciences, Peking University, Beijing 100871, China; bzhang@pku.edu.cn

**Keywords:** Bmp, *lmo4a*, zebrafish, inner ear development, balancing

## Abstract

Background: In vertebrates, the development of the inner ear is a delicate process, whereas its relating molecular pathways are still poorly understood. *LMO4*, an LIM domain-only transcriptional regulator, is drawing an increasing amount of interest for its multiple roles regarding human embryonic development and the modulation of ototoxic side effects of cisplatin including cochlear apoptosis and hearing loss. The aim of the present study is to further explore the role of *lmo4a* in zebrafish inner ear development and thus explore its functional role. Methods: The Spatial Transcript Omics DataBase was referred to in order to evaluate the expression of *lmo4a* during the first 24 h of zebrafish development. In situ hybridization was applied to validate and extend the expression profile of *lmo4a* to 3 days post-fertilization. The morpholino (MO) knockdown and CRISPR/Cas9 knockout (KO) of *lmo4a* was applied. Morphological analyses of otic vesical, hair cells, statoacoustic ganglion and semicircular canals were conducted. The swimming pattern of *lmo4a* KO and MO zebrafish was tracked. In situ hybridization was further applied to verify the expression of genes of the related pathways. Rescue of the phenotype was attempted by blockage of the bmp pathway via heat shock and injection of Dorsomorphin. Results: *lmo4a* is constitutively expressed in the otic placode and otic vesicle during the early stages of zebrafish development. Knockdown and knockout of *lmo4a* both induced smaller otocysts, less hair cells, immature statoacoustic ganglion and malformed semicircular canals. Abnormal swimming patterns could be observed in both *lmo4a* MO and KO zebrafish. *eya1* in preplacodal ectoderm patterning was downregulated. *bmp2* and *bmp4* expressions were found to be upregulated and extended in *lmo4a* morphants, and blockage of the Bmp pathway partially rescued the vestibular defects. Conclusions: We concluded that *lmo4a* holds a regulative effect on the Bmp pathway and is required for the normal development of zebrafish inner ear. Our study pointed out the conservatism of *LMO4* in inner ear development between mammals and zebrafish as well as shed more light on the molecular mechanisms behind it. Further research is needed to distinguish the relationships between *lmo4* and the Bmp pathway, which may lead to diagnostic and therapeutic approaches towards human inner ear malformation.

## 1. Introduction

The vertebrate inner ear comprising a complex labyrinth of epithelial cells surrounded by a bony capsule is a tiny organ with a delicate partition and division of functions, whereby the cochlea participates in hearing and the vestibule provides a sense of balance [1]. The molecular mechanisms underlying the development of this complex organ from the simple otic placode have been a focus of research for decades. While the process is reported to involve numerous pathways, including Wnt, Shh and Bmp [2], the overall mechanisms remain largely unknown at present.

A review by Ma et al. [3] documented crucial roles of the Bmp signaling pathway in the development of the inner ear from as early as the induction of the placode to the subsequent patterning of the vestibule and cochlea as well as functional maturation. In zebrafish, *bmp2b* is expressed as early as 18–20 h post-fertilization. At the otocyst stage, expression patterns of *bmp2b* and *bmp4* in zebrafish are similar to that of *Bmp4* mRNA across multiple species including Xenopus, mouse and chick [4,5,6,7]. The expression pattern of *Bmp4* mRNA in the chick is correspondent to all eight chicken sensory organs, demonstrating its patterning effect in the inner ear [6].

Lim-domain only 4 (*LMO4* [MIM: 603129]) has been characterized as an LIM domain-only transcriptional regulator. Following its identification in 1998 [8], *LMO4* was shown to be closely associated with breast neoplasia [9]. In 2006, differential expression of Lmo genes in the developing mouse inner ear was first reported. Furthermore, regionalized expression patterns of *Lmo4* were closely linked to morphogenesis of the inner ear [10]. Targeted disruption of *Lmo4* in the mouse model led to dysmorphogenesis of the vestibule and negative regulation of sensory organ formation in the mammalian cochlea [11,12]. In addition to its role in inner ear development, recent research has revealed that *Lmo4* and its downstream proteins play vital roles in mediating the ototoxicity of cisplatin, including cochlear apoptosis and hearing loss [13,14,15,16,17,18]. *Lmo4* is proposed to act under the regulation of *Bmp4* [19], with more recent evidence showing that *Bmp2* regulates *Lmo4* via an intermediate gene, *Ntn1* [20]. However, the precise regulatory interactions between *Lmo4* and *Bmp2* remain inconclusive, and research has yet to point out the regulative effect of *Lmo4* back on the expression of *Bmps*. The majority of research on *Lmo4* to date has been limited to rodent models. Further studies on other animal models are required to investigate the structural and functional conservation of the gene.

In the current study, we aimed to illustrate the expression and function of *lmo4a* in zebrafish’s early development stages and thus explore its role in the normal development of hearing. Here, we examined the spatiotemporal developmental expression profile of *lmo4a* in otic organs in zebrafish and performed the knockdown and knockout of the gene. The role of *lmo4a* in the process of otic and vestibular development was evaluated, and its relationship with *bmp4* was validated as well. Our collective data suggest that *lmo4a* not only participates in patterning of the preplacodal ectoderm that leads to formation of the otic placode but also helps regulate the appropriate level and distribution of bmp during compartmentalization of the vestibular apparatus.

## 2. Materials and Methods

### 2.1. Animals

All animal experiments were approved by the Institutional Animal Care and Use Committee (IACUC) of Peking University (LSC-LiuD-01). Wild-type Tubingen strain and transgenic zebrafish lines were maintained and bred using standard zebrafish husbandry methods [21]. Zebrafish embryos were obtained from pairings of TU wild-type adult fish and maintained in E2 embryo medium (15.0 mM NaCl, 0.5 mM KCl, 1.0 mM MgSO_4_, 0.15 mM KH_2_PO_4_, 0.05 mM Na_2_HPO_4_, 1.0 mM CaCl_2_, 0.7 mM NaHCO_3_). Environmental conditions: light–dark cycle, 14 h–10 h. Water temperature was controlled at 28.5 ± 0.5 °C.

We employed Tg (hsp70l:dnBmpr-GFP)w30 (tBR) transgenic zebrafish [22] and activated heat shock-inducible transgenes by incubating embryos at 38.5 °C for 40 min at 18 hpf (hours post-fertilization). Transgenes were sorted according to GFP expression and raised until fixation. To prevent pigment formation, embryos older than 28 hpf were treated with 0.0045% 1-phenyl-2-thiourea (PTU). Before treatment with Dorsomorphin (see “Small-molecule chemical inhibition”) at 22 hpf, embryos were anaesthetized with tricaine (Sigma-Aldrich, St. Louis, MO, USA) and immobilized in 3% methylcellulose.

### 2.2. The Use of Spatial Transcript Omics DataBase

The expression of *lmo4a* in the early stages of zebrafish embryonic development was examined in the Spatial Transcript Omics DataBase (STOmics DB, Shenzhen, China) (https://db.cngb.org/stomics/, accessed on 5 July 2022) [23]. The expression of *lmo4a* was examined in both the single-cell RNA (scRNA) sequencing and the stereo-sequencing (stereoseq) data. Bin_annotation was applied for the analyzation of stereoseq data.

### 2.3. In Situ Hybridization and Immunohistochemistry

RNA probes were transcribed from linearized plasmids using proper polymerase (Roche Diagnostics, Tokyo, Japan; Stratagene Japan, Tokyo, Japan). Whole mount in situ hybridization and immunostaining were performed according to the standard protocol of Westerfield [21]. Embryos were fixed overnight in 4% paraformaldehyde. They then went through an additional dehydration process in methanol and were stored in methanol at −20 °C before in situ hybridization, which was not needed before immunohistochemistry. The following probes were used: *lmo4a* (forward primer: 5′-CCTGACCCTGAAAGCCTTAG-3′, reverse primer 5′-TTGGCAGACACTGAATAGCA-3′), *eya1* [24], *bmp2b* and *bmp4* [25], antibodies used included anti-myo6, anti-Hu (Invitrogen, Carlsbad, CA, USA), anti-phospho-histone 3 (anti-PH3) (Sigma-Aldrich, St. Louis, MO, USA), anti-BrdU (Developmental Studies Hybridoma Bank, Iowa City, IA, USA), and anti-GFP (1:500; rabbit or mouse, Invitrogen, USA).

### 2.4. Phalloidin Staining

Embryos were fixed overnight in 4% paraformaldehyde. After fixation, embryos were rinsed in PBS and permeabilized with PBS/2%Triton-X100 for 4 h at room temperature. Following rinsing in PBS, embryos were incubated overnight in 2.5 g/mL FITC-phalloidin (Sigma) and washed in PBS before use in the experiments.

### 2.5. Small-Molecule Chemical Inhibition

To efficiently block Bmp signaling, embryos were treated by injecting small-molecule inhibitors into otic vesicles at 22 hpf. Dorsomorphin (DM) (Sigma, P5499) was diluted in DMSO/0.5% Phenol Red at a concentration of 200 μM and 1 nL was injected into the lateral projection of one ear. Control injections were performed by injecting DMSO/0.5% Phenol Red without Dorsomorphin. Aliquots of DM stock solution were indefinitely frozen at −80 °C without detectable loss of activity. Repeated freeze–thaw cycles led to a significant reduction in DM activity. Once thawed, aliquots could be stored for several days at 4 °C.

### 2.6. Antisense Morpholino Oligonucleotides and mRNA Misexpression

Using the sequences of this clone and zebrafish genome, we designed an antisense oligonucleotide (MO) targeting the boundary between intron 2 and exon 3 of *lmo4a* pre-mRNA (*lmo4a* MO: 5′- ACACTGGAGGAGAAAAACCAAGCGA-3′). MO (Gene Tools, LLC, Philomath, OR, USA) was resuspended in water, and 8 ng MO was co-injected into one-cell stage embryos with 0.05% Phenol red as described earlier [26]. To examine the efficiency of *lmo4a* MO, RT-PCR was performed using whole-embryo RNA from 20 embryos at 24 hpf with the following primers to detect spliced and unspliced *lmo4a* mRNA transcripts: 5′-TCTATTCCTGCCAGTGAG-3′ and 5′-ATCTGAGGGAGTGCGTAT-3′. In vitro mRNA synthesis was performed using a specific RNA synthesis kit (Ambion, Austin, TX, USA) and 200 pg purified *lmo4a* mRNA was injected into embryos at the 2-cell stage.

### 2.7. Zebrafish lmo4a Knockout

The CRISPR/Cas9 system was utilized to knock out *lmo4a* in zebrafish. A rapid method for directed gene knockout in G0 zebrafish was used in accordance with previous work [27]. The guide RNAs (gRNAs) were designed prior to construction of the Genome-Scale Guide Set using the CRISPR Design Website (http://chopchop.cbu.uib.no/, accessed on 14 September 2020). The gRNA productions, using the pMD19-gata5_gRNA scaffold as the template, were generated by annealing and elongating a forward primer containing a T7 promoter, the guide sequence, the standard chimeric gRNA scaffold (5′-GTT TTA GAG CTA GAA ATA GC-3′) and a standard reverse primer tracr rev (5′-AAA AAA AGC ACC GAC TCG GTG CCA C-3′). The gRNA-related sequences used are presented in Table 1. Four gRNAs were mixed and co-injected with Cas9 protein into 1-cell stage embryos. The efficacy of gRNA was verified by extracting crude genomic DNA from the Tuebingen zebrafish embryos (used as the control group) and the whole *lmo4a*-knockout zebrafish embryos, followed by PCR amplification and sequencing.

### 2.8. Microscopy and Image Analysis

Embryos in 70% glycerol were mounted on glass slides. Fluorescent images were obtained using a Leica TCS-NT confocal microscope (Leica, Heerbrugg, Switzerland) attached to a Leica DM RBE microscope or a Leica SP5-AOBS confocal microscope attached to a Leica DM I6000 microscope. Confocal images were processed using ImageJ (NIH, Bethesda, MD, USA). Bright-field images were obtained under a Zeiss Axiophot microscope (Zeiss, Munich, Germany). All images were assembled with Adobe Photoshop. All panels are lateral views with the anterior towards the left and dorsal towards the top, unless indicated otherwise.

Cell counting and the measurement of length or area were carried out either manually under microscope or with ImageJ. For example, we manually counted the hair cell number in representative cross sections of the utricle and saccule aided by myo6 and Phalloidin staining. The measurement of the largest diameter of the otocyst and the largest area of the statoacoustic ganglion (SAG) on axial cross sections were performed using ImageJ.

### 2.9. Motion Trajectory Measurement

Motion trajectory acquisition and analysis of the zebrafish were performed using DanioVision (Noldus Information Technology BV, Wageningen, The Netherlands), a zebrafish behavioral acquisition system. To a 12-well plate, 8 wells were added with 1 fish per well and the trajectories were acquired after 30 min of acclimation in the detection chamber. After the initial 30-min darkness, the routine for the light stimulation was applied three times for 10 s each, followed by darkness for 10 min. The trajectory, velocity and rotation of the movements were recorded using EthoVision XT10 software (Noldus Information Technology BV, Wageningen, The Netherlands).

### 2.10. Statistical Analysis

Data are presented as mean ± standard errors of measurement (SEM) and statistical significance was assessed with *t*-test using GraphPad Prism 6.0 (GraphPad Software Inc., San Diego, CA, USA), with * *p* < 0.05 considered significant. A total of 20 control embryos and 20 morphant embryos were included in the statistical analysis of hair cell numbers of utricle and saccules.

## 3. Results

### 3.1. lmo4a Is Constitutively Expressed in the Otic Placode and Otic Vesicle

In order to initially evaluate the expression of *lmo4a* in the early stages of zebrafish embryonic development, we referred to the recently published zebrafish STOmics DB (Figure 1A–H). First, the scRNA data were examined. At 5.25 h post-fertilization (hpf), *lmo4a* expression could be found in the presumptive ectoderm, endoderm and mesoderm (Figure 1A). At 10 and 12 hpf, however, *lmo4a* was concentrated in the endoderm (Figure 1B,C). A concentrated expression of *lmo4a* could be found in the otic placode and otic vesicle at 18 and 24 hpf, respectively (Figure 1D,E). Then, the stereoseq data of 6 time points (3, 5.25, 10, 12, 18, 24 hpf) were examined (Figure 1F). In accordance with the scRNA, concentrated expression of *lmo4a* could be found in the otic vesicle at 18 and 24 hpf (Figure 1G,H).

In order to verify the abovementioned information as well as broaden the time window of the expression, whole-mount in situ hybridization was performed during the first 5 days of embryonic development. The *lmo4a* transcript was detected as early as the ovum stage, indicative of maternal expression (Figure 1I). At 6 hpf, *lmo4a* was mainly expressed in the endoderm, mesoderm (non-axial) (Figure 1K), which was in accordance with the STOmics data. The gene was detected in the ventrolateral and medial portions of newly formed otocysts at 18 hpf (Figure 1L,M). At 48 hpf, *lmo4a* expression areas were likely to include the domain of newly emerging hair cells, detected in the anterior, ventrolateral and posterior domains, corresponding to the positions of three developing cristae (Figure 1N,O) [28]. The expression profile of *lmo4a* in zebrafish resembled that of the corresponding protein in mouse [12]. The consistent and regionalized expression of *lmo4a* in the otic placode and otic vesicle supports its role in inner ear development.

### 3.2. Targeted Knockdown and Knockout of lmo4a 

In view of the finding that the expression of *lmo4a* was relatively concentrated in the inner ear, we further investigated its role in inner ear development. To this end, an antisense morpholino oligonucleotide (MO) was designed to inhibit splicing of *lmo4a* pre-mRNA, which subsequently resulted in dysfunction of the gene (Figure 2A,B). Knockdown efficiency was confirmed via amplification with RT-PCR using whole-embryo RNA from 20 embryos at 24 hpf (Figure 2C), followed by recovery and sequencing of amplified products. Despite incomplete disruption of splicing, stable phenotypes were induced. Four gRNAs were co-injected with Cas9 into one-cell embryos. Evaluation of *lmo4a* knockout at 1 dpf showed that the four gRNAs had efficacies of 83.4%, 81.8%, 67.9% and 27.5% (Figure 2D–G).

### 3.3. Loss of lmo4a Results in Deficiency of Sensory Organs in Zebrafish Otocysts

Compared with the control embryos, the *lmo4a* morphant embryos showed defective development of otic vesicle with smaller otocysts and thicker otic epithelia (Figure 3A–D). To confirm the association between *lmo4a* expression and auditory components, hair cells and SAG were labeled using immunofluorescence imaging (Figure 3E–H). Lesser hair cells and smaller, immature SAG were observed in *lmo4a* morphants. To further verify our observation, we performed statistical analyses of the hair cell number and area of SAG using 20 embryos of WT and *lmo4a* morphant each (Figure 3I–K). Insufficient *lmo4a* induced a 30% and 25% decrease in hair cell number in the utricle and saccule at 36 hpf, respectively (Figure 3I,J). The SAG area of the morphant embryos was reduced by 33%, compared to the control (Figure 3K). These differences were statistically significant (*n* = 20 for both control and *lmo4a* MO group), clearly indicating overall disruption and delay of the process.

### 3.4. Knockdown of lmo4a Expression Leads to Defects in Semicircular Canal Morphogenesis

In addition to malformation of otocysts, hair cells and statoacoustic ganglion, initiation of semicircular canal formation was abnormal (Figure 3L–Q). As shown in Figure 3L, at 48 hpf, the anterior, posterior and lateral projections should have begun to protrude into the otic lumen. However, no protrusions were visible in the *lmo4a* morphants (Figure 3O). At 55 hpf, the anterior, posterior and lateral protrusions were fused while ventral protrusions grew out in the control (Figure 3M) while only the lateral protrusions were visible in the *lmo4a* morphants (Figure 3P). At 3 dpf, hubs of semicircular canals in the inner ears of the control larvae exhibited a regular shape (Figure 3N). In contrast, the *lmo4a* morphants still only contained lateral protrusions (Figure 3Q).

### 3.5. Knockout of lmo4a Replicates the Phenotypic Changes in lmo4a Knockdown and Leads to Abnormal Swimming Behavior of Zebrafish

In order to further validate the phenotypic changes in *lmo4a* morphants, we constructed *lmo4a* knockout zebrafish and embryos were cultured up to 15 dpf. The *lmo4a* knockout embryos replicated the defective development of otic vesicle with smaller otocysts and thicker otic epithelia observed in *lmo4a* morphants (Figure 4A–H). Statistical analyses demonstrated a significant difference between the largest diameters of the WT and *lmo4a* KO embryos at all time points tested (Figure 4I).

Since the inner ear plays crucial roles in zebrafish hearing and balancing, the obvious inner ear defects led us to further investigate the swimming behavior of the zebrafish in order to evaluate if the balancing function was impaired by *lmo4a* KO. Notably, the wild-type animals usually swam or rested with their back upwards at 3 dpf, while the *lmo4a* morphants always rested with the lateral side upwards and showed a deficiency in startle response. We tracked the swimming path of the zebrafish at 15 dpf (Figure 4J). The movement of the *lmo4a* KO zebrafish was more active compared with that of the WT group. The frequencies of both clockwise rotation and counterclockwise rotation were significantly higher in the KO group (Figure 4K,L).

### 3.6. Overall Otic Deformities Result from Disruption of Preplacodal Ectoderm (PPE) Patterning

Further to phenotypic changes, we investigated whether *lmo4a* influences the overall otic induction process. The genes eya1 have been established as critical participants in placode induction [29,30,31]. In situ hybridization at the bud and 3-somite stages revealed the downregulation of these genes in *lmo4a* morphants (Figure 5A,B), indicating an overall decrease in gene expression in caudal PPE and PPE-derived cells, which led to a marked disruption of PPE patterning.

### 3.7. bmp2b and bmp4 Expressions Are Upregulated and Extended in lmo4a Morphants

Based on the significance of the Bmp signaling pathway in the development of the inner ear and the finding that lmo4 acts downstream of *Bmp2b* and *Bmp4* [3,19,20], we evaluated whether *lmo4a* is reciprocally associated with expression of *bmp2b* and *bmp4* (Figure 5C–F). Examination of expression patterns of *bmp2b* and *bmp4* in both the control and *lmo4a* morphants revealed a distinct upregulation and extended area of Bmp signaling. In the control embryos at 28 hpf, *bmp4* was expressed strongly at the anterior and posterior ends and weakly in the lateral and dorsal regions of the vesicle (Figure 5C). However, in the *lmo4a* morphants, ectopic expression of *bmp4* in the dorsal region was observed and the original anterior and posterior regions of expression were extended dorsally at 28 hpf (Figure 5D). At 36 hpf, transcripts of *bmp2b* were no longer in accordance with positions of cristae, instead showing dispersed distribution throughout the vesicle (Figure 5E,F).

### 3.8. Blockage of Bmp Signaling Attenuates Vestibular Defects Caused by Downregulation of lmo4a in Zebrafish

The above results clearly demonstrate an association of upregulation and extended expression of Bmps with the defective phenotypes of morphants, such as immature protrusions and failure of semicircular canal formation. The Bmp antagonist, chordin (chd) morpholino, at a concentration of 500 μM × 1 nL, could replicate the *lmo4a* MO phenotype with immature protrusions to an extent (Figure 5G,H). The collective findings indicate that regulated levels and restricted expression areas of Bmp signaling are crucial for the development of protrusions and semicircular canals.

Rescue attempts were additionally performed to establish the causal relationship between excessive Bmp signaling and semicircular defects. Heat shock of tBR (a transgenic line containing a truncated Type I Bmp receptor) under control of a heat shock promoter [22] at 22 hpf successfully rescued the semicircular canal phenotype of lmo4a MO (Figure 5I,J). Rescue of the phenotype was achieved to a lower extent upon suppression of Bmp signaling through injection of DM (an AMP-activated protein kinase inhibitor) into morphant otic vesicles at 22 hpf (Figure 5K,L). Based on these findings, we suggest that elevated Bmp activity inhibits protrusion and semicircular canal formation and lmo4a plays a major role in this process through the modulation of Bmp levels and expression domains.

## 4. Discussion

The molecular mechanisms underlying the development of the vertebrate inner ear are yet to be clarified. In this study, using zebrafish as a model, we demonstrated that *lmo4a* is required for otic organ formation from early preplacode induction until vestibular development. Transparent zebrafish embryos facilitate direct observation of not only developmental defects but also expression patterns of *lmo4a*, along with its impact on other genes. Our experiments have disclosed regulatory effects of *lmo4a* on *bmp2b* and *bmp4* for the first time, providing novel insights into the Bmp signaling pathway. Furthermore, the structural and functional conservation of *lmo4a* between zebrafish and mice supports the utility of zebrafish as a model for future studies on this gene.

### 4.1. lmo4a Contributes to Inner Ear Sensory Organ Formation through Regulating PPE Patterning

Maternal expression of *lmo4a* at the ovum stage indicates an important function in overall embryonic development, which is, to a degree, consistent with the fatal phenotypes previously reported in *Lmo4*-null mice [32]. At 14 hpf, the otic placode was first recognized as an oval grouping of cells within the ectoderm and *lmo4a* expressed in the ventrolateral and media regions. In pace with the rapid conversion of placode into a hollow vesicle, the expression areas of *lmo4a* included newly emerged hair cells and anterior, ventrolateral and posterior cristae. As the sensory patches are closely associated with the ventral portion [33], the discovery of sustained *lmo4a* mRNA expression in the developing inner ear has led to the hypothesis that this transcription factor potentially functions in the induction or differentiation of inner ear sensory epithelia.

Indeed, the targeted knockdown of *lmo4a* by morpholino induced a series of inner ear developmental abnormalities, including smaller otocysts and thicker otic epithelia, loss of 30% and 25% hair cells in the utricle and saccule, and 33% decrease in immature SAG area, with the latter two showing statistical significance. The analogous reduction in hair cell ratio and SAG suggest that inner ear dysplasia is potentially due to a deficiency of early otic specification. In genetic analyses, the expression patterns of PPE markers, such as eya1, were significantly altered. These results indicate that *lmo4a* is involved in triggering a delay in the overall otic induction process as early as PPE patterning, which could explain the otic vesicle defects. Moreover, apoptosis and proliferation did not affect early otic compartmentalization.

### 4.2. lmo4a Plays a Regulatory Role in Semicircular Canal Morphogenesis

Semicircular canals in zebrafish are located in the dorsal region of the inner ear and share identical morphology. Although details of the early stages of semicircular canal morphogenesis differ among vertebrate species, the dorsal otic part is relatively conserved. The normal development of semicircular canals in zebrafish ear involves the outgrowth of epithelial protrusions, contact recognition, adhesion and formation of fusion plates, and formation of pillars and independent semicircular canals. Our experiments showed that *lmo4a* is expressed in outgrowing projections of zebrafish ear and is required for correct plate fusion and pillar formation. Loss of *lmo4a* resulted in complete ablation of the three semicircular canals. Complete rescue of these features was achieved by injecting *lmo4a* mRNA at the two-cell stage in *lmo4a* morphants, strongly supporting a regulatory role of the gene in semicircular canal morphogenesis (Figure 5M–P). Accordingly, we conclude that *lmo4a* not only participates in early developmental events, such as PPE patterning, but also later events, such as semicircular canal morphogenesis.

### 4.3. lmo4 Regulates Bmp Signaling during Regulation of Semicircular Canal Formation

During morphogenesis of the vestibular apparatus, the outgrowth of semicircular canals is driven by BMPs, in particular, *Bmp4* and *Bmp2b* [19,28,34]. Research on the complex processes underlying normal patterning of the vestibular apparatus has shown that deletion of several important factors, including Wnt (signaling from the dorsal hindbrain), Dlx and Hmx (members of two homeobox-containing gene families within the inner ear) result in disorganization or absence of Bmp expression within the affected presumptive cristae, highlighting a central role of the Bmp pathway in vestibular structure formation [35,36,37,38].

In *lmo4a* morphant ear, a marked upregulation and extended area of expression of *bmp2b* and *bmp4* were observed, suggesting a regulatory effect of *lmo4a* on these genes. *Lmo4* is proposed to act under regulation of *Bmp4* [19]. Furthermore, recent research has shown that *Bmp2* regulates *Lmo4* via an intermediate gene, *Ntn1* [20]. However, data from the present study highlight a more complex relationship between *Bmps* and *Lmo4*. Based on the assumption that *Lmo4* acts downstream of *Bmps*, it is reasonable to propose a reciprocal regulatory effect of *Lmo4* on *Bmps* in the form of a negative feedback loop (Figure 5A). DM (AMP-activated protein kinase inhibitor) treatment of *lmo4a* morphants rescued the formation of anterior protrusions and heat shock of tBR at 22 hpf in *lmo4a* morphants attenuated Bmp activity, which rescued the formation of anterior, posterior, ventral protrusions as well as fusion of anterior and lateral protrusions. These data support a causal relationship between excessive Bmp signaling and semicircular defects and further demonstrate that the lack of *Lmo4* can be bypassed.

In view of the earlier and current findings, we propose a mechanistic model in which (1) *Bmp2b* and *Bmp4* act upstream of *Lmo4*; (2) *Lmo4* exerts a negative feedback reciprocal regulatory effect on *Bmp2b* and *Bmp4*; (3) regulated levels and restricted expression domains of *Lmo4* and Bmps are crucial for the accurate formation of protrusions and semicircular canals (Figure 6A).

However, the possibility that *Lmo4* serves as a regulator of Bmps (Figure 6B) or that *Lmo4* and Bmps are involved in two parallel signaling pathways with mutual regulatory interactions (Figure 6C) cannot be ruled out.

Several details remain to be clarified in our model. For instance, although *Lmo4*-null mice displayed similar phenotypes as our morphants, *Bmp4* was markedly downregulated [12] in contrast to our findings. We propose that both different degrees of disruption by either conditional knockout or morpholino-induced depletion and interspecific heterogeneity could contribute to this phenomenon. The detailed molecular mechanisms, structural and functional conservation among species and means by which these molecules regulate the otic developmental process require further exploration. In addition, a previous study demonstrated that Lmo4 reciprocally interacts with Lmx1a in the regulation of inner ear formation [39]. Our findings added more to the understanding of Lmo4 in the development of the inner ear and highlight the importance of the regulatory effect of *Lmo4* on Bmps, which would be potentially beneficial for the understanding of inner ear development and for the development of therapeutic strategies of related diseases.

### 4.4. Conservation of the Function of lmo4 between Species

In addition to our current study of *lmo4a* function in zebrafish, previous researchers have studied the function of the homologous genes in mammalians including mouse [10,11,12]. These researchers demonstrated that in *Lmo4a*-null otocysts, outpouches of semicircular canals failed to form and cell proliferation was reduced, indicating that *Lmo4* is essential for the development of semicircular canals and their associated sensory cristae [12]. These findings were basically consistent with our observations in zebrafish. However, previous researchers also demonstrated the formation of an ectopic organ of Corti (eOC) located in the lateral cochlea in *Lmo4a*-null mouse otocysts, which was not observed in either our *lmo4a* MO or KO zebrafish [11]. Collectively, these observations indicate that the function of *lmo4* was basically conservative with slight differences among species. Future studies were needed to further explore the function of *lmo4* in zebrafish on a cellular level.

### 4.5. Clinical Significance of LMO4

In humans, *LMO4* is known for its association with breast neoplasia and other types of carcinoma [9,35,36,37]. Apart from a series of studies by Jamesdaniel and colleagues, which demonstrated that *LMO4* and its downstream targets mediate the ototoxic effects of the commonly used anti-neoplastic agent cisplatin [13,14,16,17,40], its role in overall embryonic and inner ear development has not been a particular focus of clinical attention until now. We believe that this gene holds significant potential in inner ear development based on current research using rodent and fish models as well as clinical genetic analysis. Ten entries concerning the CNVs of fragments encompassing the *LMO4* region on Chr1 have been retrieved from the NCBI ClinVar database, the majority showing different degrees of developmental delay. Intriguingly, the development of ear abnormalities was reported in a patient with a variation ID of 442387 [41]. These clinical features clearly highlight the potential implication of *LMO4* in human inner ear development. Further clinical exploration of the developmental roles of *LMO4* should provide insights that facilitate advancements in effective genetic diagnostic and therapeutic approaches for human inner ear malformations.

### 4.6. Limitations of the Current Study

In our current study, the expression and function of *lmo4a* in zebrafish inner ear and vestibular development were examined. An expression profile was established, phenotypic anomalies were observed in both *lmo4a* morphants and in *lmo4a* knockout zebrafish, and the mechanisms lying behind were briefly illustrated. However, due to time limitations, the current study did not establish a stable *lmo4a* KO fish line to further validate the phenotype. Moreover, the detailed mechanism between *lmo4a* and the Bmp pathway remains to be clarified. We are currently constructing the *lmo4a* KO fish line and we hope future studies will further elucidate the molecular function of the gene.

## 5. Conclusions

Based on the collective findings, we conclude that *lmo4a* is required for the normal development of zebrafish inner ear, in particular, inner ear sensory organ formation and semicircular canal morphogenesis. The gene affects sensory organ formation through the regulation of PPE patterning and its role in semicircular canal formation is based on a reciprocal regulatory interaction with the Bmp pathway. Notably, the *lmo4a* gene is conserved to an extent between mammals and zebrafish. The abovementioned findings highlight the potential regulatory roles of the gene on normal hearing development. Further studies are warranted to elucidate the molecular mechanisms of inner ear development and yield innovative novel diagnostic and therapeutic strategies for human inner ear malformations.

## Figures and Tables

**Figure 1 genes-14-01371-f001:**
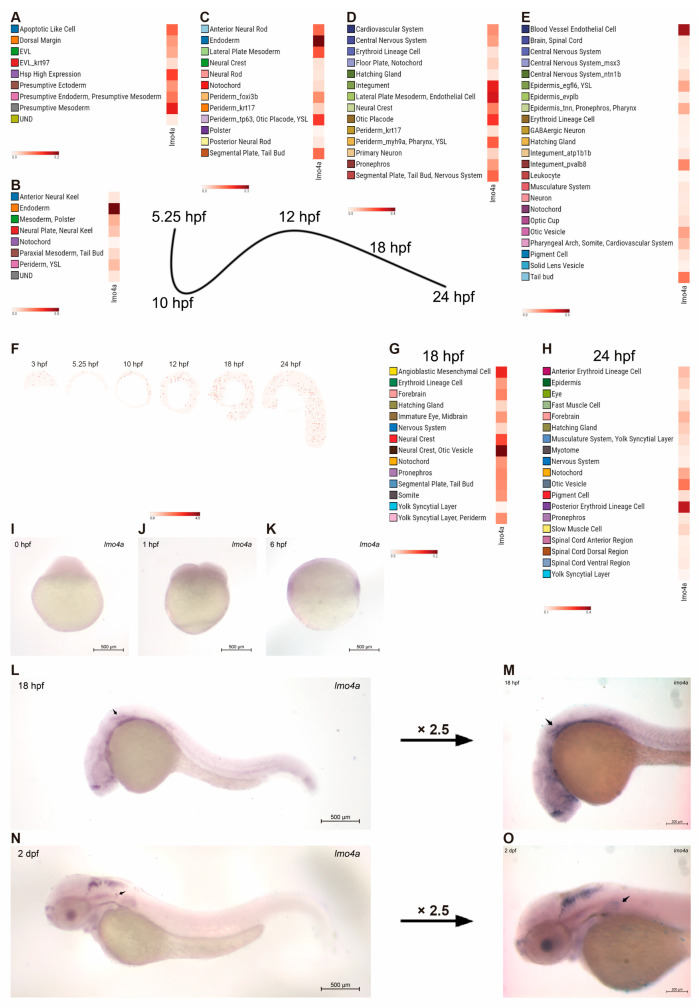
**Spatiotemporal expression of *lmo4a* during the first 2 days post-fertilization.** The spatiotemporal expression of *lmo4a* was examined by using both STOmics DB (**A**–**H**) and in situ hybridization (**I**–**N**). (**A**–**E**) Single-cell RNA sequencing data showed concentrated expression of *lmo4a* in otic placode and otic vesicle at 18 and 24 hpf. (**F**–**H**) Stereo sequencing showed concentrated expression of *lmo4a* in otic vesicle at 18 and 24 hpf. (**I**,**J**) Maternal *lmo4a* expression is visible at the ovum stage and 1 hpf. (**K**) Gastrula period showing *lmo4a* expression. (**L**,**M**) Lateral view at 18 hpf shows restricted expression in the somatic mesoderm and otic vesicle. (**N**,**O**) Lateral view at 48 hpf shows *lmo4a* concentrated expression in the hindbrain and otic vesicle.

**Figure 2 genes-14-01371-f002:**
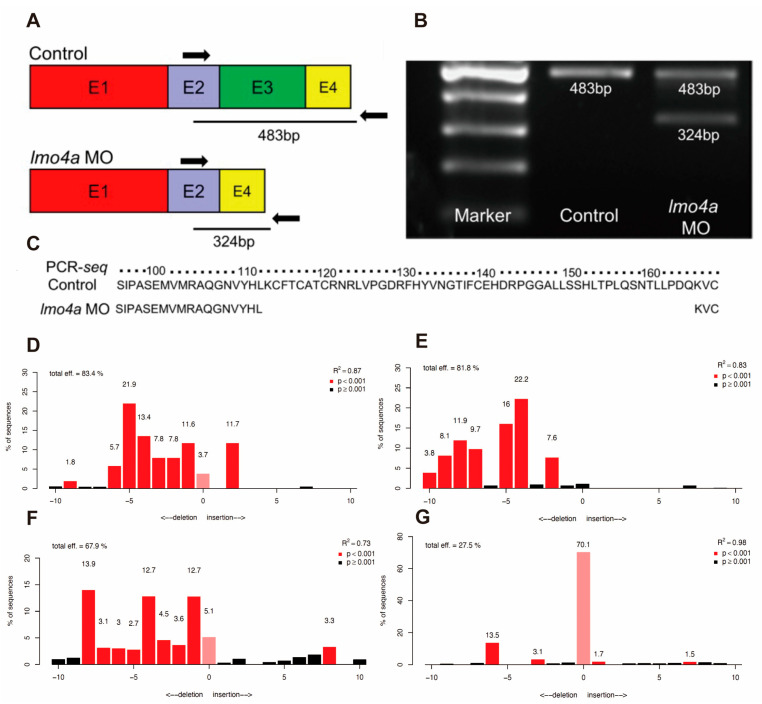
**Morpholino-targeted knockdown and targeted knockout of *lmo4a*.** (**A**) Bands of 483 bp and 324 bp represent correctly spliced and artificially mis-spliced *lmo4a* mRNA. Colored boxes indicate exons. Arrows signify primers used for RT-PCR and the black bar depicts *lmo4a* MO. (**B**) Protein alignment including the truncated protein generated by *lmo4a* MO. (**C**) RT-PCR of control embryos and those injected with 2 pg *lmo4a* MO showing effective but incomplete disruption of splicing (m, 100 bp ladder marker). (**D**–**G**) PCR and Sequencing analysis showing the *lmo4a* knockout efficiencies of the four gRNA targets.

**Figure 3 genes-14-01371-f003:**
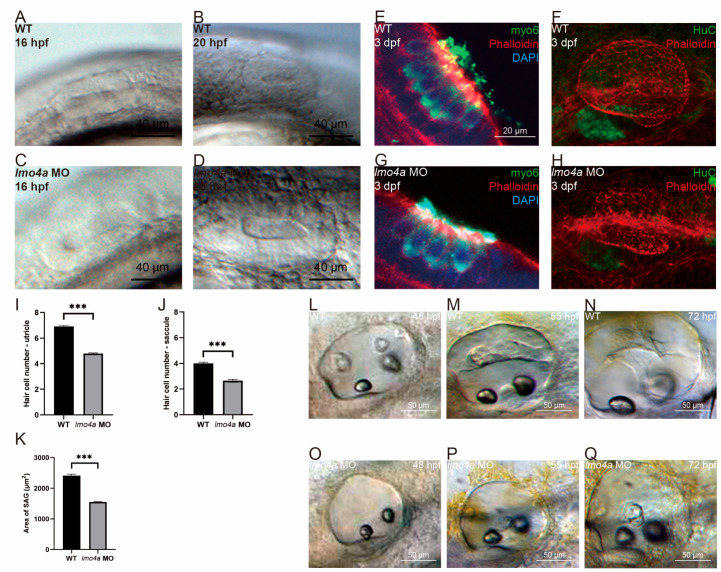
**Detailed phenotypic and behavioral changes in *lmo4a* morphants.** All panels represent lateral views of ears. (**A**–**D**) Otic morphology of *lmo4a* morphants at different stages displaying smaller otocysts and thicker otic epithelia compared to WT. (**E**,**G**) Hair cell number was reduced in *lmo4a* morphants, as determined from triple labeling with phalloidin (red, staining the cell framework and stereocilium), anti-myo6 (green, uniformly distributed throughout hair cell cytoplasm) and DAPI (blue, labeling the cell nucleus). (**F**,**H**) Embryos were double-labeled with phalloidin and anti-HuC (green, staining SAG), which revealed an evident reduction in the area of SAG in *lmo4a* morphants. (**I**,**J**) Statistical analysis of hair cell numbers of utricle and saccules between control and *lmo4a* morphants showed significant differences (*** *p* < 0.001, *n* = 20). (**K**) Statistical analysis of the area of SAG between control and *lmo4a* morphants showed significant differences (*** *p* < 0.001, *n* = 20). (**L**) Anterior, posterior and lateral protrusions were observed in control larvae at 48 hpf; (**M**) Anterior, posterior and lateral protrusions were fused while ventral protrusions grew out in the control group at 55 hpf. (**N**) The hubs of semicircular canals exhibited regular shapes in the control group at 72 hpf. (**O**) No protrusions were visible in *lmo4a* morphants at 48 hpf. (**P**) Only lateral protrusions were visible in *lmo4a* morphants at 55 hpf. (**Q**) Only lateral protrusions were visible in *lmo4a* morphants at 72 hpf.

**Figure 4 genes-14-01371-f004:**
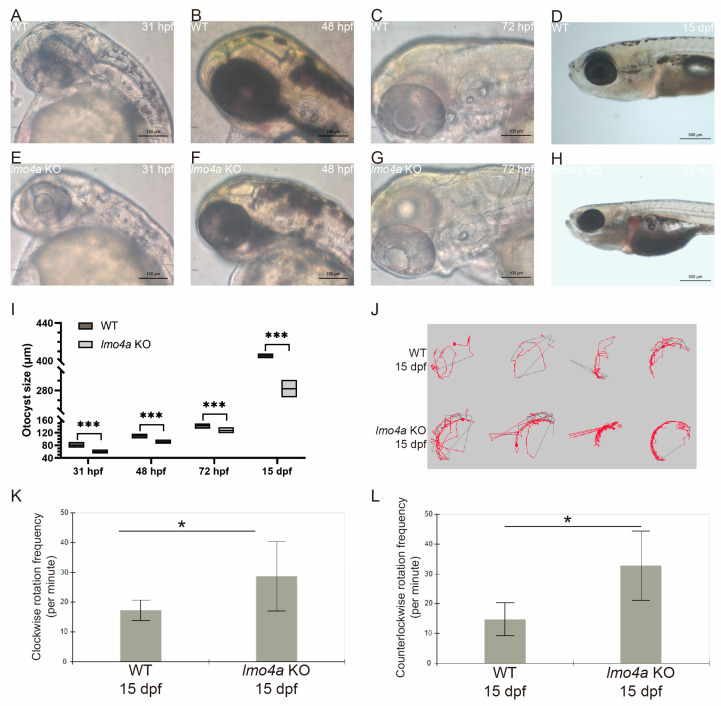
Phenotypic replication and behavioral tests of *lmo4a* KO embryos. All panels of photos represent lateral views of ears. (**A**–**H**) Otic morphology of *lmo4a* KO embryos at different stages displaying smaller otocysts and thicker otic epithelia compared to WT. (**I**) Statistical analysis confirmed that the otocyst size of *lmo4a* KO group was significantly smaller at 31, 48, 72 hpf and 15 dpf (*** *p* < 0.001, *n* = 20). (**J**) Swimming paths of WT and *lmo4a* KO zebrafish were tracked, showing a more active pattern in the KO group. (**K**,**L**) The frequencies of both clockwise rotation and counterclockwise rotation were significantly higher in the KO group (* *p* < 0.05, *n* = 4).

**Figure 5 genes-14-01371-f005:**
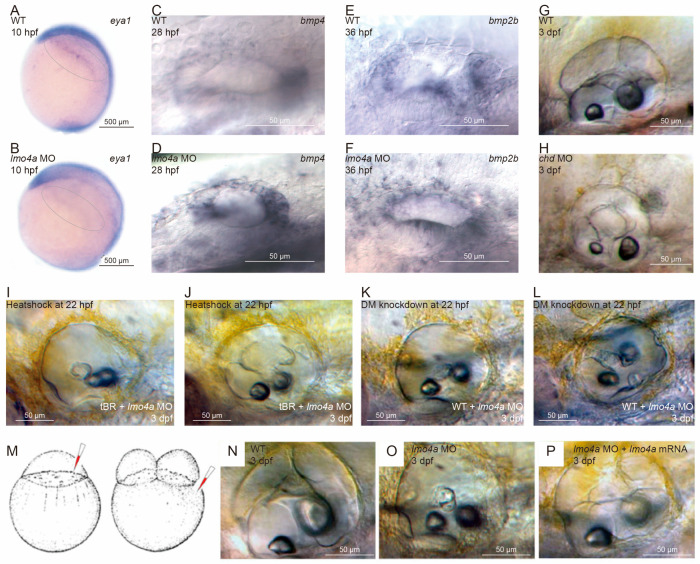
Expression patterns of marker genes of PPE and bmps in *lmo4a* morphants and rescue of deficiencies in *lmo4a* morphants via blockage of Bmp signaling. All panels represent lateral views. (**A**) Expressions of eya1 in WT embryos at 10 hpf. (**B**) Expressions of eya1 in *lmo4a* MO embryos were decreased. (**C**) *Bmp4* was expressed strongly at the anterior and posterior ends but weakly in the lateral and dorsal regions of the vesicle in the control group at 28 hpf. (**D**) Ectopic expression of *bmp4* was observed in the dorsal region, with extended original anterior and posterior regions of expression in *lmo4a* morphants at 28 hpf. (**E**) *bmp2b* was expressed strongly at the anterior, posterior and ventral regions in the control group at 36 hpf. (**F**) Dispersed distribution of *bmp2b* throughout the vesicle in *lmo4a* morphants at 36 hpf. (**G**,**H**) Chordin (chd) morpholino could replicate the *lmo4a* MO phenotype with immature protrusions to an extent. (**I**,**J**) Heat shock of tBR at 22 hpf successfully rescued the semicircular canal phenotype of *lmo4a* MO. (**K**,**L**) DM knockdown of Bmp at 22 hpf achieved rescue, but to a lower extent. (**M**–**P**) Rescue was achieved by injecting *lmo4a* mRNA at the two-cell stage in *lmo4a* morphants.

**Figure 6 genes-14-01371-f006:**
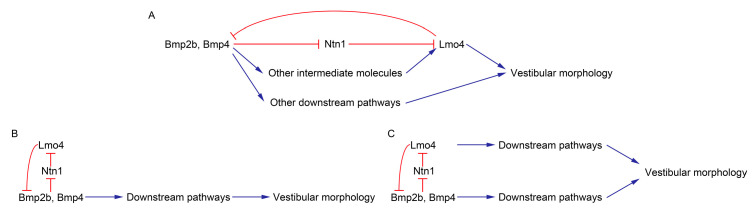
Potential molecular mechanisms linking Bmps and *lmo4a.* (**A**) *Lmo4* acts downstream of the Bmp signaling pathway. (**B**) *Lmo4* acts as a regulator of Bmps. (**C**) *Lmo4* and Bmps function in two parallel pathways with mutual regulatory interactions.

**Table 1 genes-14-01371-t001:** The gRNA-related sequences.

Target	Sequence(5′ → 3′)
*lmo4a* Target 1	TAATACGACTCACTATAGGCTCCGCGGTTGCGGTAACGTTTTAGAGCTAGAAATAGC
*lmo4a* Target 2	TAATACGACTCACTATAGGCTGCTCTTCTCTATGGATGTTTTAGAGCTAGAAATAGC
*lmo4a* Target 3	TAATACGACTCACTATAGGACCTGCTTCAGCAAAGGAGTTTTAGAGCTAGAAATAGC
*lmo4a* Target 4	TAATACGACTCACTATAGGCCGAAATCGCTTGGTTCCGTTTTAGAGCTAGAAATAGC

## Data Availability

All data generated or analyzed during this study are included in this published article.

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
