# Peer review of "lmo4a Contributes to Zebrafish Inner Ear and Vestibular Development via Regulation of the Bmp Pathway"

_genes, 2023, doi:10.3390/genes14071371_

Round 1

Reviewer 1 Report

Using knockdown and knockout zebrafish for the lmo4 gene, Sun et al., have studied the implication of the lmo4 gene in the zebrafish inner ear development. The authors also analyzed the interaction between the lmo4a gene and the Bmp pathway during inner ear development. The results showed that lmo4a is required for normal inner ear development, mainly for the semicircle channels.

Line 26, 126 – DM what that means. The authors should use write the full word before using abbreviations.

Line 188: “corresponding to the positions of three developing cristae (Figure 1. M-N)”. A higher magnification picture is needed to see the gene expression in the cristae.

The scale bar in Figure 1 is missing.

Line201:  preiodàperiod

Line 206: “lmo4a is highly expressed in the inner ear” I will say that lmo4a expression is weak in the inner ear.

Line 224: “the largest diameters of WT and lmo4a KO embryos at all time points tested (Figure 3. I)” It is not clear if the authors are measuring the inner ear diameter or the embryo diameter.

Figure 3A-H: Scale bars are not visible.

Figure 3J-V: Scale bars are missing.

Figure 4: The scale bars and the number of samples used are missing.

Figure 4A-B: an arrow indicating the region of interest will help to compare between WT and lmo4a knockdown.

Line 301: It is important for the paper to show a picture of the chordin (chd) morpholino embryo showing a similar phenotype that lmo4a embryos.

Line 369-371: “Complete rescue of these features was achieved by injecting lmo4a mRNA at the two-cell stage in lmo4a morphants, strongly supporting a regulatory role of the gene in semicircular canal morphogenesis.”       These results are not included in this study.

Reviewer 2 Report

In this manuscript, Sun et al, described the role of Lmo4a during zebrasfish inner ear development using morpholino and KO models. They identified that KOs have smaller otocyst size than WT and morphants showed decreased HC and SGN number with semicircular canal dysmorphogenesis. They further identified this phenotype is due to up-regulation of Bmps an lowering Bmp signal rescues this phenotype.

Overall, they described the phenotypes nicely and corelates those with mouse model. However, there are some concerns that need to be clarified before publication.

Major concerns

1. Mostly, they analyzed morphants rather than KOs. Based on Figure 2, it seems like morphants would not be complete. They should analyze KOs and use morphants as supportive data since morphant’s data would not completely represent loss of Lmo4a.

2. Confirmation of Lmo4a KO should be included.

3. Figure 3 is confusing. Which HCs and SGNs they image and measure?

4. Authors mentioned that malformation of auditory organ but no data supporting this.

5. Authors should clarify which function they intend to analyze for the behavior test.

6. It would be better to test whether Bmp rescue also improves behavioral deficit.

7. Some methods are missing: HCs and SGNs counting, Otocyst measurement, behavior test.

Reviewer 3 Report

Minor points:

Line 13: “Vertebrates” instead of “vertebrate”

Line 19: remove “in order to”

Line 26: what is the significance of DM?

Line 88: replace , by . just before Environmental

Line 113-114: The authors did not mention the sequences of the probes for dlx3b, six4.1, eya1, bmp2b and bmp4

Line 136: Put lmo4 in italics (lmo4)

Line 187: Add a space just before detected

Line 189: Discard the point just before the reference (25)

Line 287: bmp2b instead of bmp2

Line 288: bmp2b instead of bmp2

Lmo4 is in italics in the figure 4 and the discussion part but “normal” (results part). Check all along the manuscript. Please respect the nomenclature for zebrafish gene.

Figure 1:

Line 199: The authors indicate “(F-H) Stereo 199 sequencing showed concentrated expression of lmo4a in otic vesicle at 18 and 24 hpf”. However, in Figure 1F, 6 time points (3, 5.25, 10, 12, 18, 24 hpf) was examined (see Line 177).

1F: Add the time points directly on the figure.

1M-N: Add arrows or asterisks.

Figure 3 J, K, L, M: Write “phalloidin” instead of “philloidin”

Figure 3 X, Y: P value is indicated in the legend, but the corresponding * is not indicated on the figure.

This reference is not mentioned in the manuscript:

Rosati R, Shahab M, Ramkumar V, Jamesdaniel S. Lmo4 Deficiency Enhances Susceptibility to Cisplatin-Induced Cochlear Apoptosis and Hearing Loss. Mol Neurobiol. 2021 May;58(5):2019-2029. doi: 10.1007/s12035-020-02226-4. Epub 2021 Jan 7. PMID: 33411315; PMCID: PMC8026651.

For the discussion part, this other article is not mentioned: Huang Y, Hill J, Yatteau A, Wong L, Jiang T, Petrovic J, Gan L, Dong L, Wu DK. Reciprocal Negative Regulation Between Lmx1a and Lmo4 Is Required for Inner Ear Formation. J Neurosci. 2018 Jun 6;38(23):5429-5440. doi: 10.1523/JNEUROSCI.2484-17.2018. Epub 2018 May 16. PMID: 29769265; PMCID: PMC5990987.

Round 2

Reviewer 2 Report

This manuscript improves much.

I have couple of other comments to be addressed before publication.

Additional comments

1. I understand the time issue regarding KO study. I suggest to re-arrange the manuscript showing all MO data first and KO data last stating that KO data support phenotypes in MOs and add discussion for the future direction using KOs.

2. For hair cell and SGN counting, did you count total hair cells of the organ or certain representative area? For otocyst measurement, did you measure the longest or shortest axis? Precise description is required so that following researchers can recapitulate.

3. All data suggest loss of LMO4 in zebrafish results in vestibular organs malformation and defects. But auditory organ defects are not obvious. Small otocyst is not necessarily correlated to auditory system defects. I suggest deleting “hearing loss” from the keywords and add phenotypic difference between mouse and zebrafish at discussion.
